# Hippocampal ensemble dynamics timestamp events in long-term memory

Alon Rubin[†], Nitzan Geva[†], Liron Sheintuch, Yaniv Ziv*

Department of Neurobiology, Weizmann Institute of Science, Rehovot, Israel

**Abstract** The capacity to remember temporal relationships between different events is essential to episodic memory, but little is currently known about its underlying mechanisms. We performed time-lapse imaging of thousands of neurons over weeks in the hippocampal CA1 of mice as they repeatedly visited two distinct environments. Longitudinal analysis exposed ongoing environment-independent evolution of episodic representations, despite stable place field locations and constant remapping between the two environments. These dynamics time-stamped experienced events via neuronal ensembles that had cellular composition and activity patterns unique to specific points in time. Temporally close episodes shared a common timestamp regardless of the spatial context in which they occurred. Temporally remote episodes had distinct timestamps, even if they occurred within the same spatial context. Our results suggest that days-scale hippocampal ensemble dynamics could support the formation of a mental timeline in which experienced events could be mnemonically associated or dissociated based on their temporal distance.

*For correspondence: yaniv.ziv@
weizmann.ac.il

[†]These authors contributed
equally to this work

Competing interest: See
page 14

Reviewing editor: Howard
Eichenbaum, Boston University,
United States

The capacity to remember temporal relationships between different events is essential to our autobiographical memory and sense of self. Autobiographical or episodic memory relies on the hippocampus, whose neurons are thought to encode information about *where* and *when* events have occurred (*Davachi and DuBrow, 2015*; *Eichenbaum, 2014*; *Howard et al., 2014*; *Rolls, 2010*; *Shapiro, 2014*; *Tulving, 2002*). Hippocampal place cells encode the spatial location of an animal through localized firing patterns, and have long been considered a substrate for long-term memory of the location in which events occurred (*O'Keefe and Dostrovsky, 1971*; *O'Keefe, 1978*). Whereas ample knowledge exists regarding the encoding of location, relatively little is known about the neural mechanisms that enable the encoding of the time in which events occur. Recent work has revealed that in familiar environments hippocampal place cell activity is dynamic over timescales that range from minutes to weeks (*Howard and Kahana, 2002*; *Mankin et al., 2015*; *Mankin et al., 2012*; *Manns et al., 2007*; *Ziv et al., 2013*). For timescales that are greater than one day, these dynamics primarily result from ongoing changes in the subsets of place cells that are active during repeated visits to the same fixed environment (*Ziv et al., 2013*). Such dynamics may contribute information about the temporal relationship between events by providing a unique code that functions as a 'timestamp'. If such timestamps exist, they would likely aid long-term memory by reducing interference between traces of events that occur at different times at the same place, or that are similar in that they share contextual components such as sensory experience and behavior. Moreover, to support the formation of a mental timeline of experienced events in long-term memory, and the capacity to mentally 'time-travel' during memory recall (*Kragel et al., 2015*; *Nyberg et al., 2010*), timestamps should change gradually and continuously with time. Such gradual changes in the ensembles of place cells active during similar events on different days have been recently reported, but the extent that these dynamics actually carry temporal information remains unclear (*Mankin et al., 2012*; *Ziv et al., 2013*). We consider two alternative hypotheses regarding the possible contribution of the observed dynamics to coding of time. According to one hypothesis, the dynamics in the ensemble activity over days is unique to the environment in which it is observed,

**eLife digest** The ability to recall the timing of events is an important feature of long-term memory. Episodic memory, the mental account of "what" happened, "where" and "when", depends on a region of a brain called the hippocampus. Certain neurons in the hippocampus, called place-cells, are known to capture information about the locations an animal has visited so that a specific pattern of place cell activity marks each location an animal visits. However, it is not clear how the brain can mark the relationship between the timing of different events.

Some studies have documented gradual changes in the activity patterns of the place cells over time, which could help mark time. If these changes are specific to a particular environment then they would not allow animals to associate in memory events that occurred close in time (for instance, in the same day) if these events occurred in different environments. To do that, a certain component of the changes in the activity patterns would have to be independent of any specific environment or context in which events occur.

Now, Rubin, Geva et al. have captured time-lapse images of the activity of thousands of hippocampal cells in mice as they explored two different environments on repeated occasions over a two-week period. The environments had different shapes, textures, visual cues, and odors. The mice were allowed to explore each environment daily for more than a week prior to the time-lapse filming so that they would be very familiar with the two environments. During the filming portion of the experiments, the mice visited one environment in the morning, and then the other in the afternoon.

The analysis of the images revealed what appeared to be unique patterns of cell activity for specific days, which gradually changed over the course of the experiment. The patterns persisted even when the animals switched to a new environment during the same day, but were different for visits to the same environment on different days. Next, Rubin, Geva et al. used the patterns of activity collected from the mice while they were in one environment to create a timeline of events. From this timeline, it was possible to accurately deduce which day each visit to the other environment occurred based on the patterns of hippocampal cell activity alone. One challenge that stems from this work is to understand the biological mechanisms that drive the patterns in neuronal activity over timescales that are relevant for long-term memory.

and independent from the dynamics in other, dissimilar environments. In this case, the dynamics may contribute ordinal information about different events that occur within a given environment, but will not contribute to associations in memory between events that happen close in time if these events occurred in different or dissimilar environments. An alternative hypothesis asserts that certain aspects of the days-scale dynamics in the ensemble activity are common to different environments. Such environment-nonspecific dynamics could support a linkage in long-term memory between dissimilar events that occur at temporal proximity. If this is the case, we would expect the hippocampal representations of events that occur in different spatial environments but in temporal proximity (e.g. the same day) to share common time-varying components.

To test these alternative hypotheses we investigated hippocampal neuronal representations of different spatial contexts over multiple days and weeks. We combined head-mounted miniaturized fluorescence microscopes (*Ghosh et al., 2011*; *Ziv et al., 2013*), chronic microendoscopy (*Barretto et al., 2011*), and viral-vector based expression of a genetically encoded $Ca^{2+}$ indicator (*Chen et al., 2013*), to longitudinally image the $Ca^{2+}$ dynamics of large populations (> 1,000 per mouse) of hippocampal CA1 pyramidal cells in freely behaving mice that repeatedly explored two familiar environments (*Figure 1A*). To avoid circadian effects we alternated the two environments between AM and PM sessions, 4–5 hr apart. Each session consisted of five 3-min trials. To maximize the perceived differences between the environments, we constructed linear tracks (environments A and B) that differed in shape, floor texture, surrounding proximal and distal visual cues, odor, and flavor of the water reward (see Materials and methods). To uncover the time-dependent coding dynamics, while minimizing changes in place codes induced by learning, we familiarized the mice with the two environments before starting the experiment (*Figure 1A*). During pre-training, the mice ran on each linear track for 15 min per day for 8–11 days, until they performed at least 60 laps in

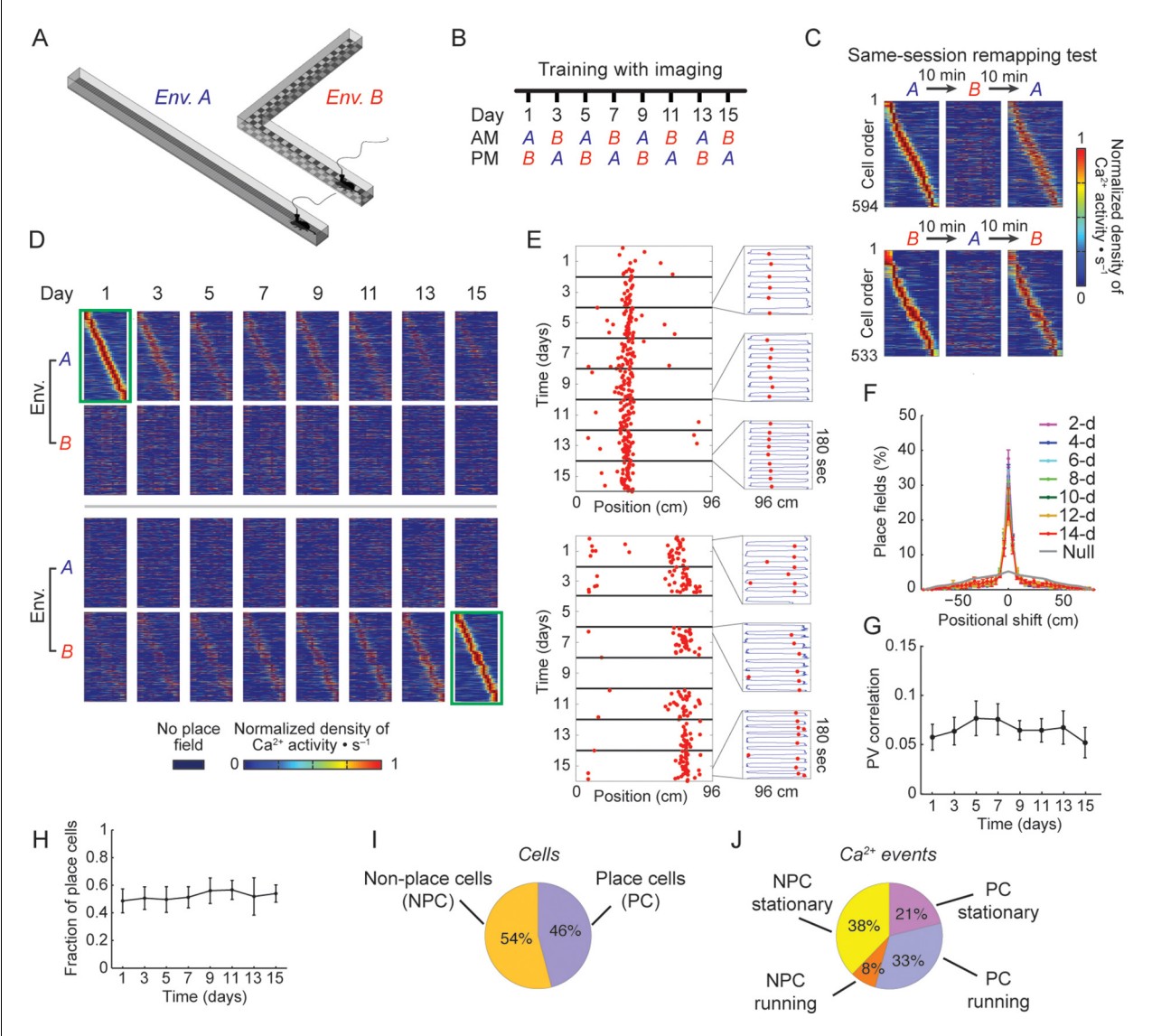

**Figure 1.** Stable spatial remapping between two familiar environments despite evolving place codes. (A) Mice trained to run back and forth and collect a liquid reward in two different linear tracks. (B) Experimental timeline. After a pre-training period, we trained and imaged mice in the two environments every two days for 15 days. Each session (AM and PM) consisted of five 3-min trials. (C) Remapping tests within the same session performed on day 16 (top panels) and day 17 (bottom panels) confirmed that CA1 representations of the two environments were different. The test consisted of two trials in one environment followed by three trials in the other environment, and then an additional three trials in the first environment (A→B→A, or B→A→B). Shown are place-field maps ordered according to the place field centroid position in the first two trials of the session in environment A (top) and environment B (bottom). (D) Place field maps for the same cells on multiple days of the experiment, ordered by the place fields' centroid positions in environment A on day 1 (green frame, top panels), or by the place fields' centroid positions in environment B on day 15 (green frame, bottom panels). The maps depict the changes in place cell activity patterns in different environments and on different days. (E,F) Place cells that were active on more than one day typically retained their place fields in subsequent sessions. (E) Examples of a place cell active on all days of the experiment in environment B (top), and a place cell active on some days of the experiment in environment A (bottom). Red dots indicate the location of the mouse along the track during cellular Ca$^{2+}$ excitation. Insets depict the mouse's trajectory (blue line) and location (red dots) during cellular Ca$^{2+}$ excitation in individual 3-min trials taken from different days. (F) The distributions of centroid shifts (color indicates the interval between sessions, averaged over five mice, mean ± s.e.m) were, centered at zero, and distinct from the null hypothesis that place cells would randomly alter their place fields (Kolmogorov-Smirnov test, $p < 10^{-14}$ for all distributions). (G) Average correlations of place cell firing patterns (i.e. the 'population vector', PV) during running epochs at different locations along the two linear tracks were low and did not change over time (One-way repeated-measures ANOVA; F<1, p = 0.44). (H) The fraction of place cells out of the population of cells active in both environments on each day also remained stable over time (One-way repeated-measures ANOVA, $F_{(2.18, 8.71)}$ = 1.97, p = 0.19). (I) Average fractions of place cells and non-place cells, and (J) average fractions of Ca$^{2+}$ events generated by place cells and non-place cells when mice were running or stationary.

*Figure 1 continued on next page*

*Figure 1 continued*

The following figure supplements are available for figure 1:

**Figure supplement 1.** CA1 neurons exhibit selective Ca2+ responses in the two environments.

**Figure supplement 2.** Accurate across-session cell identification based on correlation or distance between candidate cells.

**Figure supplement 3.** Stable neuronal function and environment-specific activity.

**Figure supplement 4.** Activity and long-term dynamics of GCaMP6s- and GCaMP6f expressing cells.

each environment for two consecutive days. Time-lapse imaging began two days after the last pre-training session, and was performed every other day for two weeks (days 1–15, *Figure 1B*). To confirm that changes in the representations of the two environments are also evident within a single session, we performed on days 16–17 a 'remapping test' by shuttling the mice back and forth between environments within the same session, keeping the interval between environments < 10 min. This test confirmed that CA1 neuronal ensembles were differentially and selectively active in the two environments (*Figure 1—figure supplement 1*), forming place field maps that were markedly different (*Figure 1C*), in agreement with past studies of place cell remapping (*Leutgeb et al., 2004*; *Muller and Kubie, 1987*).

For longitudinal analysis of the cell activity, we revalidated a previously established routine for aligning cell locations across sessions (*Ziv et al., 2013*) and applied it to our study (*Figure 1—figure supplement 2*). Consistent with a recent study using GCaMP3 (*Ziv et al., 2013*), we found that only a small portion of the cells were active on all days of the experiment (380 and 364 cells, corresponding to 7.4 and 7.2% of the cells for environment A and B respectively; in either environment ~20% of these cells were also place coding on all days). Cells that were place coding on more than one day typically had place fields at the same spatial location (*Figure 1D–F*).

Despite extensive pre-training in both environments, changes over days in the place codes that are unique to one environment or common to both environments may have resulted from a continuous refinement of the representation due to learning or familiarization (*Karlsson and Frank, 2008*; *Lever et al., 2002*). However, this is unlikely to be the case here due to several factors: the correlations between the place field maps of the two environments (*Figure 1G*), the difference in cell activity rates between the two environments (*Figure 1—figure supplement 3A*), and the distance between the cell's peak activity in each environment (*Figure 1—figure supplement 3B*) all did not change over time, indicating a stable spatial remapping. Furthermore, the fractions of cells that were place coding in both environments (*Figure 1H*), the average number of $Ca^{2+}$ events in a session, the amplitude of the $Ca^{2+}$ events, and the fraction of cells that were place coding on different days of the experiment in each environment, remained similar between environments throughout the experiment (*Figure 1—figure supplement 3C-E*). Two-photon imaging of the CA1 before and after the experiment showed no indications of cell morbidity, such as time-dependent differences in the cells' morphology or $Ca^{2+}$ indicator expression patterns (*Tian et al., 2009*) (*Figure 1—figure supplement 3F*). Our findings that the functional properties of CA1 neurons remained stable throughout the experiment, argues against the possibility that the changes in spatial representation we observed result from pathological processes (e.g. due to overexpression of the $Ca^{2+}$ indicator or photo-toxicity). Also, the number of sessions in which cells were active, and the fraction of cells with place fields for left and right directions of the linear tracks remained similar for both environments and GCaMP6 variants (*Figure 1—figure supplement 4A,B*). Overall, our finding of stable remapping despite fluctuating place cell ensembles on different days of the experiment are consistent with previous reports of place cell remapping between different environments within the same day (*Leutgeb et al., 2004*; *Muller and Kubie, 1987*), and within a single fixed environment across days and weeks (*Ziv et al., 2013*).

In our experiment, place cells constituted 46% of the neurons active in either of the environments, and their activity during locomotion contributed 33% of the $Ca^{2+}$ transients we recorded (*Figure 1I, J*). Non-place cells generated almost half of the $Ca^{2+}$ transients (46%), mostly during resting or

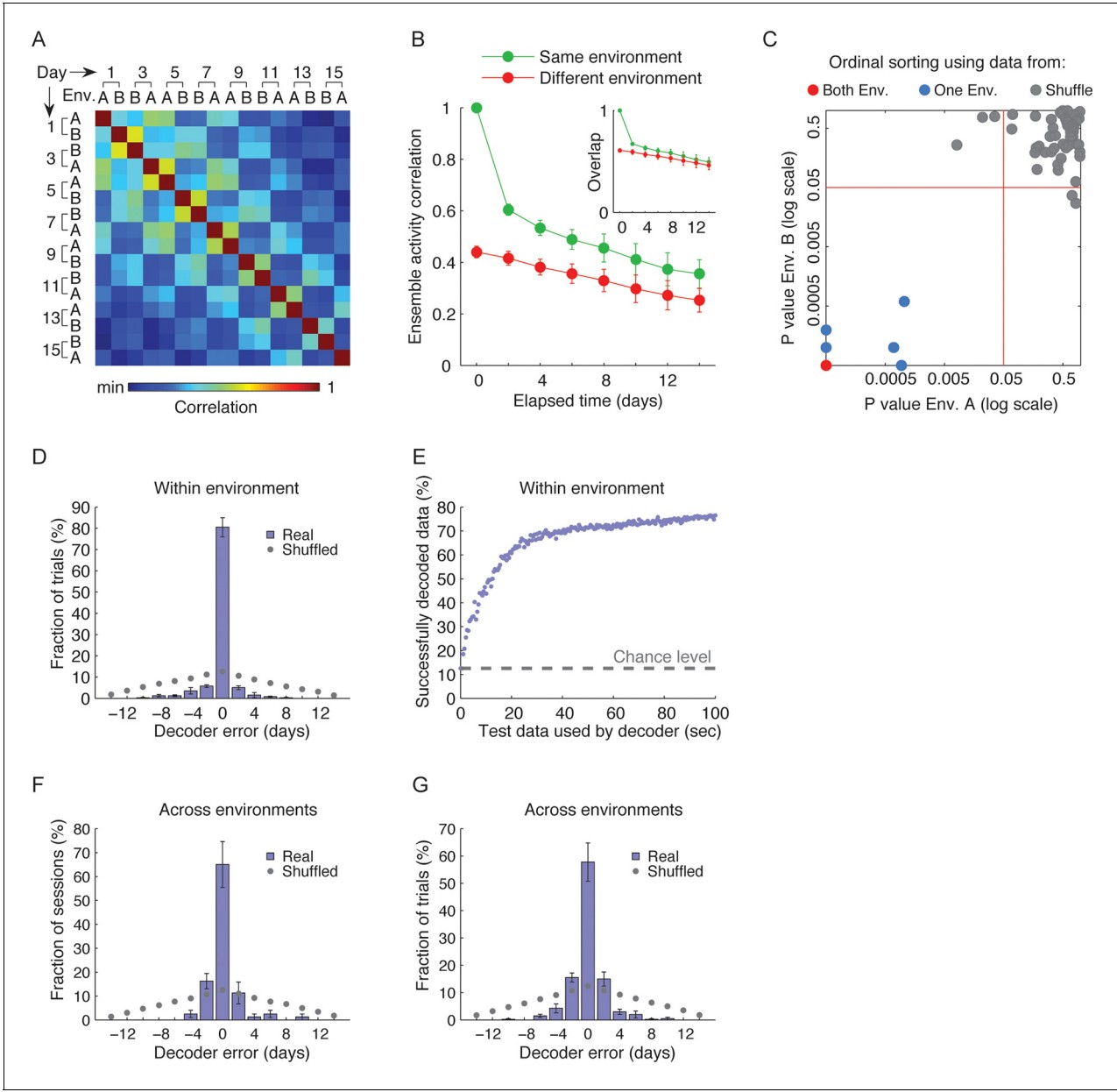

**Figure 2.** CA1 ensemble codes are informative about the temporal relationship between experienced events. (**A**) Pairwise correlations between the complete activity patterns of all cells in different environments and different days (an average of n = 5 mice). The checkerboard pattern reveals the resemblance between the episodic representations of the same environments at different times. (**B**) Correlations between cell activity patterns within the same environment (green) or between different environments (red) decay with time. This trend is also visible in **A** when considering the changes in correlations along a row. Inset, the probability that a cell active on one day will be active on subsequent days in the same environment (green) or in a different environment (red) decayed with time (mean ± s.e.m). (**C**) Performance of the ordinal time decoder. Red dots indicate perfect performance, as was achieved by using data from both environments together (see *Figure 2—figure supplement 2*). Sorting of the sessions using data from one of the environments was highly significant in comparison with the results obtained using data in which the day labels of each cell's activity patterns were randomly shuffled (gray dots, n =10 shuffles per mouse). Red lines indicate p = 0.05. (**D,E**) A 'within-environment time decoder' was trained and tested on data from the same environment. (**D**) Distributions of the decoder's errors in inferring the day from which the test data (single trials) were taken. The decoder successfully inferred the day from which the test data was taken in the majority of cases, but performed at chance level on shuffled data (gray dots, shuffled data as in **C**). (**E**) Successful time decoding (decoder error = 0 days) depended on the amount of test data used by the decoder. Shown are percentages of accurate decoding for test data of different durations. Performance using test data segments as short as 1 sec exceeded chance level (12.5%, gray horizontal line). (**F,G**) An 'across-environments time decoder' was trained on data from one environment (A or B) and tested on data from the other environment (B or A, respectively). (**F**) Distributions of the decoder's errors in inferring the day from which a test data consisting of single

*Figure 2 continued on next page*

*Figure 2 continued*

sessions (**F**) or single trials (**G**) were taken. The decoder successfully inferred the day from which the test data was taken in the majority of cases, but performed at chance level on shuffled data (shuffled data as in **C**). Data in B and D-G are mean ± s.e.m.

The following figure supplements are available for figure 2:

**Figure supplement 1.** Within-environment cell-level dynamics (A–E).
**Figure supplement 2.** Time decoders accurately expose temporal information in all individual mice.
**Figure supplement 3.** Time decoders accurately infer the time of the episode even when the tested session is excluded from the training data (A–D).
**Figure supplement 4.** CA1 ensemble place codes and non-place codes are informative about the temporal relationship between experienced events.
**Figure supplement 5.** Time decoding is robust to changes in the across-session cell identification method.
**Figure supplement 6.** The two environments can be distinguished according to CA1 place cell firing patterns regardless of recording day (A,B).

reward consumption at the ends of the linear tracks. Thus, neuronal activity during an episode on the linear track was much richer than the part of the data that is limited to place codes. Furthermore, the same cells can be place coding in one environment but active and not place coding in another, or alternatively place coding on one day and active but not place coding on other days. Thus, we sought in our subsequent analyses to account for the entire episodic representations. We therefore opted to include in the following analyses all the active cells and their Ca$^{2+}$ events, regardless of their spatial tuning properties.

We quantified the correlations between the ensemble cell activity patterns in different sessions, in either the same or different environments for pairs of sessions separated by different intervals (*Figure 2A,B*). For the same environment, both the degree of overlap between populations of cells active on different days, and correlations in their activity patterns gradually declined as a function of elapsed time (*Figure 2B*, green data). This gradual decay in the correlations at the ensemble level cannot be explained by monotonic changes in event rates at the single cell level (*Figure 2—figure supplement 1A,B*). Notably, the correlations in cell activity patterns between environments, and the probability of cells recurring within the subset that was active in both environments declined with elapsed time (*Figure 2B*, red data). Thus, hippocampal representation of the two environments co-evolved with time.

To what extent can the dynamics in the representations of the two environments infer the time in which episodes occurred? To address this question while accounting for the neuronal activity at the ensemble level, we constructed three types of neuronal 'time decoders'. The decoders are based on the knowledge that the similarity between ensemble activity patterns of different episodes decays as a function of elapsed time (*Figure 2A,B*). We first quantified whether the coding dynamics are informative about the temporal relationships between different events that occurred days to weeks apart within either one or both of the environments. We used an 'ordinal-time decoder' to determine if days-scale changes in CA1 ensemble activity are informative about the order of experienced events. This decoder examines data from all days and orders them by maximizing the mean correlation between cells' firing patterns on neighboring days (*Figure 2C* and *Figure 2—figure supplement 2A*). We tested the decoder's performance on entire sessions, (each 15 min long) for each environment separately and for both environments together, and contrasted the results with those obtained using data in which the day labels of each cell were randomly shuffled. When data from only one of the environments were included in the analysis, performance was significantly superior compared with chance levels for accurate ordinal sorting and the performance of the decoder on shuffled data (*Figure 2C*, blue data points). Notably, when data from both environments were included in the analysis, the decoder performed perfectly, accurately ordering all sessions along the experiment in all five mice (*Figure 2C*, red data points, and *Figure 2—figure supplement 2A*). Thus, day-to-day differences in the activity patterns of CA1 neurons carry ordinal information. This result argues against the possibilities that the observed differences in neuronal activity between sessions are

merely the result of inherently unreliable neuronal responses, or due to fluctuations of cell excitability over minutes to hours. Such alternative explanations are unlikely to support perfect ordinal sorting over multiple days.

Days-scale changes in ensemble activity may facilitate the binding in memory of fractions of events within temporal clusters. Accordingly, we expected that different trials or fractions of trials from the same session would be similarly time-stamped. To determine the consistency of such time-stamps within the same environment, we devised a decoder that estimates the day in which an episode was recorded based on the correlations between different trials or fractions thereof ('within-environment time decoder'). Using single 3-min trials as test data, this decoder accurately inferred the days from which the test data was taken (98 ± 1% of the cases). To determine to what degree temporal information is carried in the changes in neuronal firing rates we performed the analysis again, this time relying only on cells that were active on all days. This decoder was accurate in 80% of the cases, significantly higher than chance (z-test, p < 7 · $10^{-48}$, *Figure 2D*). The decoder's performance on shuffled test data was essentially identical to chance levels (12.5% for accurate decoding). Remarkably, even data from very short, several-seconds-long fractions of trials, were sufficient for correct decoding of the day of experiment from which the test data was taken (*Figure 2E*). Thus, a unique timestamp of events arises from changes in the cell activity levels over time. Our data suggest that such timestamps are not only captured within the CA1 representations of complete episodes but are also highly consistent within short fractions of an episode.

If hippocampal dynamics timestamp different events that occur close in time, as our analysis indicates, then such an aspect of the neural code should also infer the time of an event irrespective of a specific environment or a behavioral context. To test this idea we constructed an 'across-environment time decoder' that quantifies the degree to which the time of events in one environment can be inferred by a decoder that is trained on data from another environment. We tested this decoder's ability to correctly identify from which day of the experiment the test data was taken (the test data consisted of entire sessions or single 3-min-long trials). Despite marked differences in the spatial representation of the two environments (*Figure 1*) and an inter-session interval of 4–5 hr for sessions taking place on the same day, the across-environment time decoder was able to infer the days from which the test data was taken in 65% and 58% of the cases for sessions and trials respectively, significantly higher than chance (z-test, p < 6 · $10^{-7}$ and 4 · $10^{-22}$, *Figure 2F,G*, and *Figure 2—figure supplement 2B,C*). The decoder's performance was also essentially identical to chance level when tested on data in which the day labels of the cells were randomly shuffled (*Figure 2F,G*). Note that the decoder's errors were mostly in proximal days. Indeed, when excluding the training data from the same day of the tested trials or sessions, the decoders inferred the neighboring days in the vast majority of cases (*Figure 2—figure supplement 3*). These results are consistent with our finding of a gradual co-evolution of the CA1 ensemble representation of both environments (*Figure 2A,B*). Furthermore, both place codes and non-place codes carried significant information about time (*Figure 2—figure supplement 4*), suggesting that time stamping is not intrinsically linked to specific aspects of the episodic representation such as place coding.

Here, we exposed a facet of episodic coding dynamics that has thus far remained hidden: the binding and separation of CA1 episodic codes over timescales of days–weeks and across environments. We capitalized on an imaging approach that allows the tracking of $Ca^{2+}$ dynamics in large populations of hippocampal neurons over timescales of weeks (*Ziv et al., 2013*). Our results rely on our capability to accurately register the neural data across sessions. Re-analyzing the data using a distance-based (rather than correlation based) registration routine (described in *Figure 1—figure supplement 2*) yielded highly similar results with regard to the performances of the three time decoders (*Figure 2—figure supplement 5A–E*). Furthermore, the performance of these decoders was robust to alterations in either the thresholds of correlation or distance used by each of the registration routines (*Figure 2—figure supplement 5F,G*). Overall, these independent analyses confirm that our findings do not stem from a bias intrinsic to our choice of the across-session registration method or threshold.

The ability to place different experienced events on a personal mental timeline is a central feature of episodic memory. Several recent studies have provided important insights into the hippocampal function with respect to coding of time in memory, suggesting that multiple time-coding mechanisms exist to cover different timescales and mnemonic needs (*Mankin et al., 2015*; *Mankin et al., 2012*; *Manns et al., 2007*; *Navratilova and Battaglia, 2015*). For instance, 'time cells', hippocampal

pyramidal cells that fire during a particular moment within a temporally structured episode (*Eichenbaum, 2014*; *Kraus et al., 2013*; *MacDonald et al., 2011*; *Modi et al., 2014*), may contribute to a representation of sequences of events over timescales of tens of seconds. Due to the seconds to minutes-scale nature of their activity, time cells are less likely to place individual uniquely experienced episodes on a linear temporal axis extending over days. Pyramidal neurons in the hippocampal area CA2 (upstream to CA1) were found to display representations that are similar for different contexts but highly variable upon repeated exposures over timescales of hours. It was suggested that such CA2 dynamics, when combined with spatial information from CA3, could jointly reflect spatial and temporal information in CA1 (*Mankin et al., 2012*, *2015*). However, it was argued that the rapid dynamics in CA2 could support encoding a temporal order between events over timescales that range from hours to one day (*Navratilova and Battaglia, 2015*), whereas many events in our episodic memory are ordered over much longer timescales.

Our present findings of coding dynamics over timescales of days are consistent with a key feature of episodic memory—that each experience is uniquely encoded. This allows for 'single trial' learning and events that occur close in time to be linked in long-term memory. This idea is also consistent with previous reports of changes in CA1 neuronal activity over timescales of seconds-hours (*Manns et al., 2007*), hours-days (*Mankin et al., 2012*) and days-weeks (*Ziv et al., 2013*). Such ensemble dynamics may be regarded as a form of 'remapping' in the time domain. Spatial remapping is thought to reflect a mechanism for pattern separation, which could help prevent interference between representations of similar places (*Colgin et al., 2008*). Likewise, we propose that 'remapping in time' may be a reflection of a pattern separation mechanism that could mitigate interference between memories of similar events that occurred at different times.

Another aspect of our findings that is key to episodic memory is the co-evolution of the CA1 ensemble representations across spatial environments. While previous work showed that the CA1 place code for a familiar environment changes over timescales of days–weeks (*Mankin et al., 2015*; *Mankin et al., 2012*; *Ziv et al., 2013*) it has remained unclear whether such evolution in the representation of a given environment is independent of the evolution of the representations of other environments. Our findings demonstrate that the representations of two environments were highly distinct, yet sufficiently co-evolved with time to allow us to infer the day in which events occurred in one environment using reference data from the other environment. We propose that such co-evolution may reflect a neural mechanism for binding in long-term memory events that occurred at temporal proximity, even if these events occurred in different contexts and hours-days apart.

Importantly, our data show that such binding/separation of temporally proximal/distal events does not interfere with the ability to distinguish between the two environments; although the spatial representation of each environment changed with time, within environment spatial correlations from sessions that are two weeks apart remained higher than across-environment spatial correlations from the same day (*Figure 2—figure supplement 6A,B*).

What are the mechanisms that drive ensemble dynamics and time coding over timescales of days? One potential mechanism at the network level is adult neurogenesis in the dentate gyrus (two synapses upstream to CA1) which takes place over timescales of weeks and has been suggested to support time coding via ongoing modification of the hippocampal circuitry by newborn neurons (*Aimone et al., 2006*, *2009*). Another possible mechanism is the recently found days-scale turnover of dendritic spines in basal CA1 dendrites, which could potentially drive coding dynamics at the cellular level (*Attardo et al., 2015*).

A closely related issue is the role of the interactions between the absolute passage of time and the experience itself on the observed ensemble dynamics. Previous work found no effect for the number of context exposures on the rate of change in CA1 codes over 30 hours (*Mankin et al., 2012*), but it is possible such an effect could be observed over longer timescales. Our findings are consistent with reports showing that changes in neuronal excitability govern which neurons will be recruited to code and support a memory of a given experience ('memory allocation') (*Rogerson et al., 2014*; *Yiu et al., 2014*). Repeated training episodes may alter cell excitability via consolidation or reconsolidation processes involving the expression of immediate early genes and their downstream effectors (*Dudai, 2012*; *Lee, 2008*; *Nader and Hardt, 2009*). Such processes may change episodic representations irrespective of the amount of time that has passed between repeated events. Thus, while we have shown in this study that CA1 ensemble dynamics carry temporal information over days, further work is needed to differentiate between the contributions and

functional roles of experience-dependent and experience-independent processes that take place in the brain during the passage of time.

## Materials and methods

### Animals and surgical procedures

All procedures were approved by the Weizmann Institute IACUC. Five male C57BL/6 mice aged 8-12 weeks at the start were used in this study. Mice were housed with 1-4 cage-mates in cages with running wheels, and underwent two surgical procedures under isoflurane anesthesia (1.5-2% volume). First, we injected into the CA1, 400 nL of the viral vector AAV2/5-CaMKIIα-GCaMP6s or AAV2/5-CaMKIIα-GCaMP6f (~2 × $10^{13}$ particles per ml, packed by University of North Carolina Vector Core)(*Chen et al., 2013*). Stereotatic coordinates were: -1.9 mm anterio-posterior, -1.4 mm mediolateral, -1.6 mm dorsoventral from bregma. The second surgery, which took place at least one week after the viral injection, was the implantation of a glass guide tube directly above the CA1. We used a trephine drill to remove a circular part of the skull centered posterio-lateral to the viral injection site. We removed the dura and cortex above the CA1 by suction with a 29 gauge blunt needle while constantly washing the exposed tissue with sterile PBS. We then implanted an optical guide tube with its window just dorsal to, but not within, area CA1, and sealed the space between the skull and guide tube using 1.5% agarose in PBS. The exposed areas of the skull were then sealed with Metabond (Parkell, Edgewood, NY) and dental acrylic.

### $Ca^{2+}$ imaging and behavioral setup

For time-lapse imaging in freely behaving mice using an integrated miniature fluorescence microscope (nVistaHD, Inscopix), we followed a previously established protocol (*Ziv et al., 2013*). Briefly, at least three weeks after guide tube implantation, we imaged water restricted mice under isoflurane anesthesia using a two-photon microscope (Ultima IV, Bruker, Germany), equipped with a tunable Ti: Sapphire laser (Insight, Spectra Physics, Santa Clara, CA). We inserted a 'microendoscope' consisting of a single gradient refractive index lens (0.44 pitch length, 0.47 NA, GRINtech GmbH, Germany) into the guide tube, and examined $Ca^{2+}$ indicator expression and tissue health. We selected for further imaging only those mice that exhibited homogenous GCaMP6 expression and healthy appearance of the tissue. For the selected mice, we then affixed the microendoscope within the guide tube using ultraviolet-curing adhesive (Norland, NOA81, Edmund Optics, Barrington, NJ). Next, we attached the microscope's base plate to the dental acrylic cap using light cured acrylic (Flow-It ALC, Pentron, Orange, CA). After a few days, we began training the mice to run back and forth on two elevated linear tracks (Environments A and B). Environment A was a straight 96 cm long track and Environment B was an L-shaped track consisting of two 48 cm long arms. Each environment had distinct sets of visual and tactile cues, overhead lights, flavored liquid rewards, and odor cues. Before the beginning of each pre-training or imaging session we wiped the tracks with differently scented paper towels (0.5% acetic acid for environment A and 10% ethanol for environment B). We trained the mice to run back and forth along the track by giving them a measured amount of water sweetened with commercial fruit juice concentrate, lemon flavored for track A and raspberry flavored for track B, with 2% added sugar by weight. The water reward was dispensed using a custom-made computer controlled device. To record mouse behavior, we used an overhead camera (DFK 33G445, The Imaging Source, Germany), which we synchronized with the integrated microscope. $Ca^{2+}$ imaging was performed at 20Hz. Before beginning with $Ca^{2+}$ imaging, we pre-trained the mice for 8–11 days, until the mice ran at least 60 times the entire length of each track in two consecutive days. Pre-training and imaging sessions consisted of five 3-min-long trials, with an inter-trial interval of 3 min. We imaged a total of 5 mice (two that were injected with AAV2/5-CaMKIIα-GCaMP6f and three that were injected with AAV2/5-CaMKIIα-GCaMP6s) every other day for 15 days, making for 8 recording days. Each day of the experiment consisted of two sessions (AM and PM) separated by 4–5 hr. Remapping tests within a single session were performed on days 16–17. At the end of the experiment we removed the base plate by drilling away the top acrylic cap and re-examined the health of the CA1 neurons by imaging the mice under isoflurane anesthesia using a two-photon microscope as described above.

Rubin *et al.* eLife 2015;4:e12247. DOI: 10.7554/eLife.12247                                                    9 of 16

## Processing of Ca²⁺ imaging data

We processed imaging data using commercial software (Mosaic, Inscopix) and custom MATLAB routines as previously described (*Ziv et al., 2013*). To increase computation speed, we spatially downsampled the data by a factor of two in each dimension. To correct for non-uniform illumination both in space and time, we normalized the images by dividing each pixel by the corresponding value in a smoothed version. The smoothed version was obtained by applying a Gaussian filter with a radius of 40 pixels on the videos. Normalization enhanced the appearance of the blood vessels, which were later used as stationary fiducial markers for image registration. We used rigid body image registration to correct for lateral displacements of the brain. This procedure was performed on a high contrast subregion of the normalized movies for which the blood vessels were most prominent. The registered movies were transformed to relative changes in fluorescence, $\frac{\Delta F'(t)}{F_0} = \left( F'(t) - F'_0 \right)/F'_0$, where $F'_0$ is the value for each pixel averaged over time. For the purpose of cell identification the movies were downsampled in time by a factor of five. We identified spatial filters corresponding to individual cells using an established cell-sorting algorithm that applies principal and independent component analyses (PCA and ICA) (*Mukamel et al., 2009*). For each spatial filter, we used a threshold of 50% of the filter's maximum intensity and each pixel that did not cross the threshold was set to zero. After the cells were identified, further cell sorting was performed to find the spatial filters that follow a typical cellular structure. This was done by measuring the filters' area and circularity and discarding those whose radius was smaller than 5 µm or larger than 14 µm, or which had a circularity smaller than 0.8. In some cases, the output of the PCA/ICA algorithm included more than one component that corresponded to a single cell. To eliminate such incidents, we examined all cells whose centroids were less than 18 µm apart and whenever their traces had *correlation* > 0.9, the cell with the lower average peak amplitude was discarded.

## Detection of Ca²⁺ events

Ca²⁺ activity was extracted by applying the thresholded spatial filters to the full temporal resolution (20Hz) $\Delta F'(t)/F_0$ videos. Baseline fluctuations were removed by subtracting the median trace (20 s sliding window). The Ca²⁺ traces were smoothed with a low-pass filter with a cutoff frequency of 2Hz. Ca²⁺ candidate events were detected whenever the amplitude crossed a threshold of 4 or 5 median absolute deviations (MAD), for GCaMP6s or GCaMP6f, respectively. Cellular Ca²⁺ events are characterized by fast rise and slow decay times. To capture these characteristics in our data we considered for further analysis only candidate Ca²⁺ events that followed typical indicator decay time, and decay-to-rise time ratios. In order to avoid the detection of several peaks for a single Ca²⁺ event, only peaks that were 4 or 5 MAD higher than the previous peak (within the same candidate event) and 2 or 2.5 MAD higher than the next peak for GCaMP6s or GCaMP6f, respectively, were regarded as true events. We set the Ca²⁺ event occurrence to the time of the peak fluorescence. To mitigate the effects of crosstalk (i.e., spillover of Ca²⁺ fluorescence from neighboring cells), we adopted a conservative approach, allowing only one cell of a group of neighbors (cells whose centroids are less than 18 $\mu m$ apart) to register a Ca²⁺ event in a 200 msec time window. If multiple Ca²⁺ events occurred within ~200 msec in neighboring cells, we retained only the events with highest peak $\Delta F'(t)/F_0$ value. If two neighboring cells had *correlation* > 0.9 in their events, the cell with the lower average peak amplitude was discarded.

## Registration of cells across sessions

For each session we projected centroids of all thresholded filters onto a single image. We computed the spatial cross-correlation among the projections from all sessions to align them according to a reference session. Because changing the reference did not change the alignment output, we chose the first session as the reference. This step corrected slight translations and rotation changes between sessions and yielded each cell's location in the reference coordinate system. Next, we searched for cells from different sessions that might be the same neuron. This was performed using two separate methods based on either spatial correlations or centroids distances (*Figure 1—figure supplement 2*). *Figures 1 and 2*, *Figure 1—figure supplement 3*, and *Figure 2—figure supplements 2–6* show longitudinal data for which we used the spatial correlations-based registration method. *Figure 2—figure supplement 5* shows longitudinal data for which we used the centroids distances-based registration method. Within each session, the nearest neighbors spatial correlations were always < 0.6

(*Figure 1—figure supplement 2A,C*) and the centroids distances were always > 6 $\mu m$ (*Figure 1— figure supplement 2B,D*). Between sessions, however, a large amount of cell pairs had spatial correlations > 0.6 and centroid distances < 6 $\mu m$. Pairs with spatial correlation > 0.7 or distance < 5 $\mu m$ were registered as the same neuron. In cases with more than one candidate, the cells with the minimal distance or maximal correlation were assigned to be the same neuron. Analyzing the data using a range of different thresholds demonstrated the robustness of our registration process to the choice of the threshold (*Figure 2—figure supplement 5*).

## Place fields

We analyzed mouse behavior videos using a custom MATLAB (Mathworks) routine that detected the mouse's center of mass in each frame, calculated its velocity and applied a rectangular smoothing window of 250 msec. For place field analysis, we considered periods when the mouse ran >1 cm s$^{-1}$. We divided each track into 24 bins (4 cm each) and excluded the last 2 bins at both ends of the tracks where water rewards were consumed and the mouse was generally stationary (*Ziv et al., 2013*). We computed the time spent in each bin, and the number of Ca$^{2+}$ events per bin, and smoothed these two maps ('occupancy' and 'Ca$^{2+}$ event number') using a truncated Gaussian kernel ($\sigma$ = 1.5 bins, size = 5 bins). We then computed the place field map for each neuron by dividing the two smoothed maps of Ca$^{2+}$ event number and occupancy. We separately considered place fields for left and right running directions and normalized each place field by its maximum value. We defined each place field's position at its peak value. For each place field with >5 events for a given session, we computed the spatial information (in bits per event) using the unsmoothed events-rate map of each cell, as previously described (*Markus et al., 1994*):

$$Spatial\ Information = \sum_{i} p_i(r_i/\bar{r})log_2(r_i/\bar{r})$$

Where $r_i$ is the Ca$^{2+}$ event rate of the neuron in the $i$<sup>th</sup> bin; $p_i$ is the probability of the mouse being in the $i$<sup>th</sup> bin (time spent in $i$<sup>th</sup> bin/total session time); $\bar{r}$ is the overall mean Ca$^{2+}$ event rate; and $i$ running over all the bins. We then performed 1000 distinct shuffles of animal locations during Ca$^{2+}$ events, accounting for the spatial coverage statistics at the relevant session and direction, and calculated the spatial information for each shuffle. This yielded the p value of the measured information relative to the shuffles. Place fields with p $\leq$ 0.05 were considered significant.

## Population vector correlation

To determine the level of similarity between representations of the different environments, we calculated the mean population vector correlation between them (*Leutgeb et al., 2005*). For each spatial bin (excluding the last 2 bins at both ends of the tracks) we defined the population vector as the mean event rate for each neuron given that bin's occupancy. We computed the correlation between the population vector in one environment with that of the matching location in the other environment, and averaged the scores over all positions. Since there are two edges to each of the two linear tracks there are two possible transformations between them. Therefore, we used the one that resulted in higher global population vector correlation.

## Statistical analysis

We generated the null hypothesis for place fields' displacements between a pair of days by taking the measured centers of place fields in the same environment on the two days and shuffling cells' identities on each of the days. We calculated the distribution of all displacements and averaged them over 10,000 distinct pairs of shuffles. *Figure 1F* shows the mean null hypothesis for the displacement curve found by averaging over all pairs of days for a given elapsed time. For the analyses shown in *Figure 1G,H*, and *Figure 1—figure supplement 3A-E* we used analysis of variance (ANOVA) with repeated measures. Greenhouse-Geisser estimates of sphericity were used for degrees of freedom adjustment.

## Time decoders

To capture the temporal information encoded in the hippocampal neural representations of different episodes we constructed three types of time decoders: (1) ordinal time decoder, (2) within-

environment time decoder, and (3) across environments time decoder. The time decoders estimated the true order of the recording days from sets of eight episodes (sessions or trials) from the different days in the experiment. Decoding analyses were performed separately for five mice. Vectors of ensemble activity patterns were constructed where each element corresponded to the total number of events of one neuron within an episode. We notated the full-session ensemble activity pattern in day $d$ in environment $E$ as $V_d^E$, and the ensemble activity pattern in the $t^{th}$ trial on day $d$ in environment $E$ as $v_{d,t}^E$. To quantify the similarities between ensemble activity patterns of different episodes we calculated the Pearson correlation between the activity vectors. For the within and between environments time decoders, we normalized each correlation value between a test-data pattern and a training-data pattern by subtracting the average correlation of the training-data vector over all the vectors.

## Ordinal time decoder

To sort shuffled, unlabeled activity patterns according to their ordinal positions (chronological order), we sought to maximize the average of all correlations between activity patterns taken on neighboring days in the shuffled test-data. This was achieved by calculating the average correlation between activity patterns for all 20,160 possible permutations of the eight recording days (8!/2) (see *Figure 2—figure supplement 2A* and *Figure 2—figure supplement 5B*). The decoder output the day-ordering that maximized the average correlation:

$$\widehat{O}\big(\{V_i^E\}_{i=1:8}\big) = \underset{<d_1,d_2,d_3...d_8>}{\mathrm{argmax}}\left\{\frac{1}{7}\sum_{j=1}^{7} corr(V_{d_j}^E, V_{d_{j+1}}^E)\right\}$$

Where $\widehat{O}(\{V_i^E\}_{i=1:8})$ is the inferred days order for a set of eight ensemble activity patterns in environment $E$, $<d_1,d_2,d_3...d_8>$ is a possible ordering of the eight patterns, and $V_i^E$ is the ensemble activity pattern of the $i^{th}$ full-session in environment $E$ (environment A, environment B, or both environments together).

To obtain the significance of the ordinal time decoder performance versus chance level, we divided the number of permutations with a mean correlation that is equal or greater than the mean correlation for the correct order by the total number of possible days' permutations.

## Within environment time decoder

Within environment time decoder inferred the time in which episodes (single trials) in one environment were recorded by comparing their ensemble activity patterns to the activity patterns in all the sessions from the same environment. Specifically, we calculated the normalized correlations between single-trial ensemble activity patterns in one environment (test-data) and each of the full-session activity patterns in all the sessions from the same environment (training-data). To evade bias, in addition to the exclusion of test trials from their sessions in the training-data, we also excluded the corresponding trials from the rest of the sessions in the training-data. Then, the decoder output the time of the session that maximized the correlation with the test-data:

$$\widehat{d}(v_{i,j}^E) = \underset{d}{\mathrm{argmax}}\left\{corr(v_{i,j}^E, V_d^E - v_{d,j}^E) - \underset{d'}{\mathrm{E}}\left[corr(v_{d',j}^E, V_d^E - v_{d,j}^E)\right]\right\}$$

Where $\widehat{d}(x)$ is the inferred day for ensemble activity pattern $x$, $\mathrm{E}_{d'}[\cdot]$ is the average over all $d'$, $v_{i,j}^E$ is the ensemble activity pattern in the $j^{th}$ trial of the $i^{th}$ day in environment $E$ and $V_i^E$ is the ensemble activity pattern in the $i^{th}$ full-session in environment $E$. For the results presented in *Figure 2—figure supplement 3* and *4* we applied the same decoder while excluding from the training-data the session from the day of the test-data.

## Across environments time decoder

Across environments time decoder inferred the time in which episodes (trials or sessions) in one environment were recorded, by comparing their ensemble activity patterns to the activity patterns in all the sessions in the other environment. Decoding was done at the trial level as for the within environment time decoder. For across environments time decoder, decoding was done at the session level as well. Specifically, we calculated the normalized correlation between the full-session ensemble

activity pattern in one environment (test-data) and the full-session activity patterns in all the sessions in the other environment (training-data). Then, the decoder output the time of the session that maximized the correlation with the test-data:

Trial-based:

$$\widehat{d}(v_{i,j}^{E_1}) = \underset{d}{\operatorname{argmax}} \left\{ corr(v_{i,j}^{E_1} V_d^{E_2} - v_{d,j}^{E_2}) - \underset{d'}{\mathrm{E}} \left[ corr(v_{d',j}^{E_1} V_d^{E_2} - v_{d,j}^{E_2}) \right] \right\}$$

*Session-based:*

$$\widehat{d}(V_i^{E_1}) = \underset{d}{\operatorname{argmax}} \left\{ corr(V_i^{E_1}, V_d^{E_2}) - \underset{d'}{\mathrm{E}} \left[ corr(V_{d'}^{E_1}, V_d^{E_2}) \right] \right\}$$

Where $\widehat{d}(x)$ is the inferred day for ensemble activity pattern $x$, $\mathrm{E}_{d'}[\cdot]$ is the average over all $d'$, $v_{i,j}^E$ is the ensemble activity pattern in the $j^{th}$ trial of the $i^{th}$ day in environment *E* and $V_i^E$ is the ensemble activity pattern in the $i^{th}$ full-session in environment *E*. For the results presented in *Figure 2—figure supplement 3* we applied the same decoder while excluding from the training-data the session from the day of the test-data.

## Measures of divergence

We used two measures of divergence between the representations of the two environments: 'activity divergence' and 'peak displacement'.

### Activity divergence

The ratio of the average absolute difference in event rate between trials in different environments divided by the average absolute difference in average event rate between trials in the same environment:

$$Activity\ Divergence(d) = \frac{<|r_{n,d,t}^E - r_{n,d,t'}^{E'}|>_{n,t,t'}^D}{<|r_{n,d,t}^E - r_{n,d,t'}^E|>_{n,t,t'}^S}$$

where $r_{n,d,t}^E$ is the event rate of the $n^{th}$ cell in the $t^{th}$ trial of the $d^{th}$ day in environment *E*, $<\cdot>_{n,t,t'}^D$ denotes the average over all cells and over all pairs of trials ($t$ and $t'$) from different environments ($E$ and $E'$), $<\cdot>_{n,t,t'}^S$ denotes the average over all cells and over all pairs of trials from the same environment and |x| denotes the absolute value of x. We modified this analysis from (*Lever et al., 2002*) to be suited for Ca$^{2+}$ imaging data.

### Peak displacement

The average distance between locations of peak activity in different environments:

$$Peak\ Displacement(d) = <|p_{n,d}^E - p_{n,d}^{E'}|>_n$$

where $p_{n,d}^E$ is the location of the peak event rate of the $n^{th}$ cell in the $d^{th}$ day in environment $E$, $<\cdot>_n$ denotes the average over all place cells from different environments ($E$ and $E'$), and |x| denotes the absolute value of x (distance for 1-dimnesional environments). Note that there are two possible transformations between the two environments and here we used the one that resulted in higher global population vector correlation.

## Cell-level dynamics

To investigate cell-level dynamics we analyzed the changes in event rates over time for each cell. We applied these analyses separately for each of the two environments.

### Maximal monotonic sequence

For each cell we calculated the difference in event rates between consecutive recording days, resulting in a sequence of seven difference values per cell. We then found the longest sub-sequence with the same trend (either monotonic increase or decrease in event rates) and compared the distribution

of these lengths obtained from all cells to the distribution of lengths obtained from shuffled data (per-cell random permutation of the order of eight recording days).

## Monotonicity score

In order to check whether single cells exhibit monotonic behavior that is obscured by noise we derived a more relaxed definition of monotonicity. For each cell we calculated the difference between the number of day-pairs in which the later had a higher event rate, and the number of day-pairs in which the earlier had a higher event rate, normalized by the total number of day-pairs (28). This resulted in a measure that ranges from -1 for monotonically decreasing cells and +1 for monotonically increasing cells.

## Coefficient of variation (CV)

For the cells that were active in all the sessions (in one environment) we calculated the average and the standard deviation of event rates over all eight sessions in that environment. We then compared the distribution of event rates' CV for those cells to the CV distribution of a population of simulated cells with equivalent average event rates that follow a stationary Poisson model of activity.

## Population monotonicity

We aligned each cell's activity to the day of its maximal event rate and normalized the activity of all days by the maximum event rate. Consequently, the obtained activity level in day-0 is 1 for all cells. Shuffled data was obtained by a per-cell random permutation of the order of eight recording days (*Figure 2— figure supplement 1D*). To separate the effect of event rate dynamics from cell recruitment dynamics we repeated this procedure for subgroups of cells according to the maximal number of consecutive days in which they were active (2–8 days). For each subgroup we extracted the maximal segment and computed the normalized event rates only for this segment (*Figure 2— figure supplement 1E*).

## Acknowledgements

Yaniv Ziv is incumbent of the Daniel E. Koshland SR. career development chair. We thank Aia Haruvi, Linor Balilti-Turgeman, Noa Brande-Eilat, Yadin Dudai, Ofer Yizhar, and Inbal Goshen for help, advice and comments on the manuscript.

## Additional information

### Competing interests

YZ: Has ownership interests at Inscopix Inc. The other authors declare that no competing interests exist.

### Funding

| Funder | Grant reference number | Author |
| --- | --- | --- |
| Israel Science Foundation | 2184/14 | Yaniv Ziv |
| European Research Council | 638644 | Yaniv Ziv |
| Marie Curie Actions | CIG 630852 | Yaniv Ziv |
| Weizmann Institute of Science | Hymen T. Milgrom Trust | Yaniv Ziv |
| Weizmann Institute of Science | Mr. and Mrs. Mike Kahn Fund for Alzheimer's related research | Yaniv Ziv |
| Weizmann Institute of Science | Abraham and Sonia Rochlin Foundation | Yaniv Ziv |
| Weizmann Institute of Science | Harold and Faye Liss Foundation for Brain Related Research | Yaniv Ziv |

| Weizmann Institute of Science | Lulu P and David J Levidow Fund for Alzheimer's Diseases and Neuroscience Research | Yaniv Ziv |
| Weizmann Institute of Science | Mr and Mrs Mike Kahn Fund for Alzheimer's Related Research | Yaniv Ziv |

The funders had no role in study design, data collection and interpretation, or the decision to submit the work for publication.

### Author contributions

AR, YZ, Conception and design, Analysis and interpretation of data, Drafting or revising the article; NG, Conception and design, Acquisition of data, Drafting or revising the article; LS, Analysis and interpretation of data, Drafting or revising the article

### Ethics

Animal experimentation: All animal work was approved by the Weizmann Institute institutional animal care and use committee (IACUC protocol 18030515-3).

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
