## [Decision Letter]

Thank you for submitting your work entitled "Hippocampal ensemble dynamics timestamp events in long-term memory" for consideration by *eLife*. Your article has been reviewed by three peer reviewers, and the evaluation has been overseen by a Reviewing Editor (Howard Eichenbaum) and Timothy Behrens as the Senior Editor. The following individuals involved in review of your submission have agreed to reveal their identity: Stephan Leutgeb and Albert Lee (peer reviewers).

The reviewers have discussed the reviews with one another and the Reviewing editor has drafted this decision to help you prepare a revised submission.

The reviewers were generally enthusiastic about the study and the promise of the optical recording approach highlighted here. At the same time, the reviewers were concerned about the novelty of the results compared to published work by the author and others. The reviewers were clear that substantial novelty existed, but thought that the writing of the manuscript made it unclear which results were new and which could be inferred from previous data. It is essential that the authors clearly lay out what is confirmatory of existing findings with the current approach and what is qualitatively new here. The reviewers had several other major concerns and recommendations outlined below that should be addressed.

*Reviewer #1:*

The use of calcium imaging of large numbers of neurons over many days in awake and locomoting mice is potentially a very important technological contribution to the study of how populations of hippocampal neurons change over time and contribute to memory. The question though is whether or not the technological advancement evident in the current study resulted in substantive advancement of the field above and beyond the methodology. One complication in evaluating this latter question is the extent to which the authors focused on only the most recent literature and thus provided a narrow perspective of how their study fit into existing knowledge. Numerous earlier studies and models of how gradually changing neural representations could add a temporal dimension to memory are ignored. In addition, of the more recent work that is cited (e.g., Mankin et al., 2012), the authors do not clearly show how the present results differ from and thus add to the prior results (aside from the imaging technique used to obtain the data). As a result, one gets the sense that the current study reveals more about the promise of the technique rather than contributing new findings.

Reviewer #2:

The report by Rubin et al. finds that hippocampal CA1 ensembles gradually change over time such that ensemble that are recorded several days apart do not resemble each other. In contrast, they find that CA1 ensembles that are recorded on the same day in two different environments resemble each other. While the number of cells that can be recorded with calcium imaging is certainly much higher than with standard electrophysiological techniques, the reported results largely resemble those that were obtained in previous recording studies which have already reported and proposed many of the key conceptual points. Notably, closely related publications (e.g., Manns et al., 2007; Howard and Kahana, 2002, and others) are not included here and standard analysis methods (detailed comparison of firing during immobility and across the two environments) are not performed.

1) The Introduction begins with mentioning the work of others, but in the second half, predominantly cites the work from their own group (Ziv et al., 2013). Furthermore, it is not mentioned that Mankin et al. (2012 and 2014) recorded in two environments and reported correlations between the two environments, which show a similar pattern as those reported here.

2) Paragraph four. Non-place cells are presented as if it were not a well-known fact that many cells that are not place cells are active during sharp-wave ripples and in other phases of the task. In fact, many well performed electrophysiology studies report the fraction of cells that are place cells compared to those that are only observed during immobility. A much more detailed analysis of the firing patterns of non-place cells within and across environments is needed before any conclusions about their contribution to stability over time can be drawn.

3) Paragraph five. A much more detailed analysis of the correlation across environments for place cells and non-place cells is needed. It is of course expected that correlations within environments are higher than between environments but the key question is the extent of the neuronal coding difference for the two environments. From the information that is currently provided it appears that A and B are partially correlated and that each environment is more correlated with itself from at temporally proximal compared to temporally distant time points. If A and B and A and A' are correlated, is appears trivial that B and A' are also correlated, and all classifiers and conclusions seem to be based on this finding.

4) Paragraphs six and nine. These results have been shown before by others.

5) Paragraphs eleven and thirteen. Previous similar findings are not discussed.

6) Paragraph twelve. The largest fraction of change is found within 1-2 days, which is not consistent with the time course of neurogenesis. In contrast to what is reported here (i.e., constant remapping between environments), Rangel et al. (2014) found a change in the degree of remapping over the time course of neurogenesis and when manipulating neurogenesis.

7) Paragraph thirteen. While the experimental design in Mankin et al. (2012) could distinguished between effects of elapsed time and intervening experience, no such results are reported here. If this is discussed, the text should refer to previously published data on this topic.

Reviewer #3:

In this manuscript the authors build on previous work (Ziv et al., 2013) using the same technique of freely moving 1-photon imaging of hippocampal CA1 populations employed there. In the previous study they showed significant changes to the place cell representation of a single environment over ~35 days primarily due to continual changes to the subset of cells that were active (using GCaMP3 as the indicator). In particular, a cell that was active on a given day was less and less likely to be active in the same environment over several weeks of time. However, they also showed that if a cell was active at different time points within this ~35 day period that their place fields were in the same location, indicating stability of the spatial representation in that sense. Given the amount of change they observed in the representation, it is important to know to what extent this amount of change may affect how spatial memories can be recalled over the long term. One way to investigate this is to consider how the representations of two different environments could be distinguished over time, which is the experiment presented here. Here they show that while the activity levels (a graded version of the active/inactive distinction that Ziv et al. (2013) focused on) of individual cells changed over time, the representations of the environments were distinguishable from each other with a similar level of contrast at any given time point, indicating a similar magnitude of global remapping over time. They also showed that the activity levels of both environments changed gradually over time so that the amount of similarity in activity levels at different time points could be used to distinguish how far apart in time those experiences were. An important aspect of this last point was that the activity levels of the cells changed in a similar fashion for both environments, allowing them to determine when an experience occurred in a given environment by using the activity levels over time from either that or the other environment.

Long-term tracking of the activity patterns of the same set of cells is an important and understudied area. The extent to which a representation stays the same versus changes – and the manner in which it changes – over long time periods is critical for understanding memory storage and recall. There are several points in the manuscript that I believe should be addressed, but overall this is a valuable contribution to the field of hippocampal research which is also potentially relevant for memory representations in other brain areas (e.g. the amygdala).

Major comments:

1) The Abstract exclusively emphasizes the time evolution of CA1 representations independent of spatial context. The absence of a clear statement in the Abstract that important aspects of the representations of spatial contexts remain constant over time – the preserved locations of place fields (already in the Ziv et al., 2013 Abstract) and the preserved global remapping between spatial contexts (new here) – could lead to misunderstanding by readers. Therefore, the authors should modify the Abstract so that it is clear which important aspects of the representation of spatial contexts remain stable over long time periods.

2) The ability to order representations of each environment across days could be inferred from Ziv et al. (2013). What would have been the alternative to this, given the findings in this previous work? (On the other hand, the stability of global remapping across time and the co-evolution of activity levels across multiple environments is new.) The authors should delve into the details of the time evolution of single-environment representations more. For example, is the gradual evolution of activity levels over time due to half of the neurons gradually increasing their activity levels over the 2 weeks and half decreasing? Or do many neurons increase then decrease, or decrease then increase, activity over that period? Or does some substantial fraction of cells have activity levels that remain basically the same over time? What are the percentages of each kind of pattern?

3) For the within-environment decoder, it seems not surprising that the activity pattern in a trial of a day would best correlate with the activity pattern on the rest of that day. The important issue is whether the pattern evolves gradually over time versus just being noisy over time. Therefore, what the authors should do is remove the session from the day of the test trial and see if the best match from the remaining days came from a day that was closer to the actual day than chance. Similarly, this calculation should be done for the across-environments decoder (that is, eliminating the same day's data). (However, it is useful to also have the answer when the same day is included, which is the calculation they already did.)

4) An important part of the memory function of the hippocampus would be the ability to distinguish between the two spatial contexts and correctly identify them with previous experience in each of those contexts. While the authors showed intact global remapping between environments within each day, they should analyze if they can distinguish which previous environment they are recalling across days. Specifically, they should compare how the spatial population vectors of the representation of environment A on day n for n = 3, 5, […] correlate with the spatial population vectors for environment A and separately for environment B on day n = 1. Then they should repeat it for day 5, 7, […] versus day 3, and so on. (This should also be done for environment B versus environment B or A on different days.) This analysis will essentially be an across-days combination of Figure 1 and Figure 2.

5) The authors write "Thus, having shown that remapping between familiar environments is stable over time, we tried in our subsequent analyses to account for the entire episodic representations. We therefore opted to include in the following analyses all the active cells and their Ca^2+^ events, regardless of their spatial running properties." They describe in the Materials and methods that they excluded the 2 spatial bins on each end of the tracks when defining place fields, but did they then include these bins when determining the spatial population vector correlation between representations in different environments to show remapping? And I assume based on the quoted text above that they did not exclude the 2 spatial bins on each end of the tracks when defining the "ensemble activity patterns" for each time period, which was used to do their time evolution analysis? This should be clarified in the Materials and methods and main text.

6) Related to (4): They appear to have excluded the reward-related activity at the track ends in the global remapping analysis. They appear to have included this activity in the "ensemble activity pattern" vectors – which contain the relative activity of each cell in a given environment, independent of the locations at which that activity for each cell occurred – for the time ordering analysis. Assuming this is the case, the authors should re-run the analysis for the time ordering results using only the subset of cells that show place activity and, separately, all cells excluding the track ends where reward is consumed because the remaining activity would represent the theta-state associated activity patterns. This analysis would also reveal to what extent the reward-related and other activity at the track ends contributes to their time ordering results.

7) One important difference between this study and the previous one (Ziv et al., 2013) is the use of GCaMP6s and GCaMP6f here versus the previous use of GCaMP3. Both studies point out that there is significant change in the hippocampal representation over time. However, the GCaMP6 variants are more sensitive detectors of neural activity than GCaMP3, which detects strong bursts more than single spikes. Because models of memory can depend on parameter values such as how stable activity is over time, it would be very valuable for the hippocampal field to have a quantitative comparison between the key numbers from the previous GCaMP3-based study and the GCaMP6-based values here (especially since other differences in methodology are likely to be low due to overlap in the author lists). Therefore, the authors should quantitatively compare their data to Ziv et al. (2013) by adding to the current manuscript's supplementary figures the same plots corresponding to Figure 2, Figure 3B, and Figure 3C of Ziv et al. (2013). It looks like one of these (the fraction of cells that are place cells) is already shown in both (Ziv et al., 2013 Figure 2 that shows a place cell fraction of ~10-15% and this manuscript's Figure 1 shows a value of ~50%), but it's not clear that it has been computed in exactly the same way. As for Figure 3C in Ziv et al. (2013), the previous study used a binary distinction between active and inactive cells. The current manuscript's Figure 2 uses a more graded activity level measure for cells. In the inset of Figure 2, does the "overlap" plot use exactly the same definition of active and inactive cells as Ziv et al. (2013) Figure 3C? Finally, the authors should show separate plots for GCaMP6s and for 6F, which could have different detectability for events.

8) How does the data relate to Lever et al. (2002)? In that paper, it was shown that the representations of the two environments diverged over time. Here, did the representations of the two environments separate more with time in certain aspects? While the authors show that their global remapping score is stable over time, it looks like Figure 2 shows some degree of increasing divergence even though Day 1 already represents ~2 weeks of familiarization with both environments. The authors should check for this and report any change or lack of change.

---

## [Author Response]

*Reviewers were generally enthusiastic about the study and the promise of the optical recording approach highlighted here. At the same time, the reviewers were concerned about the novelty of the results compared to published work by the author and others. The reviewers were clear that substantial novelty existed, but thought that the writing of the manuscript made it unclear which results were new and which could be inferred from previous data. It is essential that the authors clearly lay out what is confirmatory of existing findings with the current approach and what is qualitatively new here. The reviewers had several other major concerns and recommendations outlined below that should be addressed.*

We thank the reviewers for their comments and suggestions for improvement of the manuscript, and appreciate that they were enthusiastic about the study. In the revised manuscript we have clarified which results are confirmatory of previous findings and which ones constitute a conceptual advance. To address specific points raised by the reviewers, we have revised two figures from the original submission and added five new Supplementary Figures, overall containing 27 new display items. We believe that these additions have substantially improved the manuscript, and we thank the reviewers for the constructive criticism that prompted these additional analyses. We summarize here the main novel contributions of our paper, as well as the new material added in the revised submission:

• We now emphasize the key novelty of our findings – that the CA1 ensemble representations of two distinct familiar environments co-evolve with time. This co-evolution allowed us to infer the day in which events occurred in one environment using reference data from the other environment, pointing to a potential mechanism for binding in long-term memory events that occurred at temporal proximity, even if these events occurred in different contexts and hours-days apart.

• Our ensemble-level analysis of the data, done for individual mice and days, and particularly our decoding approaches, provide the first quantitative measure regarding the degree to which the ensemble dynamics carry information about time.

• In the revised manuscript, we separately analyze the performance of the time decoders on place code and non-place code data, and show that both categories of activity carry information about the temporal relationship between experienced events. This finding suggests that time stamping is not intrinsically linked to specific aspects of the episodic representation such as place coding.

• We now show that the population vector correlation between activity patterns in two sessions that took place in the same environment was higher than the correlation for two sessions that took place in different environment regardless of the elapsed time between the sessions, suggesting that binding or separation of temporally proximal or distal events does not interfere with the ability to distinguish between the two environments – as expected for intact long-term memory.

• We now demonstrate that the performance of the within-environment time decoder does not solely rely on within-day similarity in activity patterns, further supporting the notion that ensemble dynamics over multiple days could associate episodes that occurred close in time.

• We now added detailed analysis, done separately for GCaMP6s and GCaMP6f, which confirms previous findings with an inferior Ca^2+^ indicator, GCaMP3.

• Finally, we show that the gradual decay in the correlations at the ensemble level between spatial representations cannot be explained by monotonic changes in event rates at the single cell level.

Reviewer #1:

*The use of calcium imaging of large numbers of neurons over many days in awake and locomoting mice is potentially a very important technological contribution to the study of how populations of hippocampal neurons change over time and contribute to memory. The question though is whether or not the technological advancement evident in the current study resulted in substantive advancement of the field above and beyond the methodology. One complication in evaluating this latter question is the extent to which the authors focused on only the most recent literature and thus provided a narrow perspective of how their study fit into existing knowledge. Numerous earlier studies and models of how gradually changing neural representations could add a temporal dimension to memory are ignored. In addition, of the more recent work that is cited (e.g., Mankin et al., 2012), the authors do not clearly show how the present results differ from and thus add to the prior results (aside from the imaging technique used to obtain the data). As a result, one gets the sense that the current study reveals more about the promise of the technique rather than contributing new findings.*

Thank you for these remarks which helped point out that we did not adequately present the conceptual novelty of this work. We hope the revised manuscript now clarifies this matter. In this work we tested the hypothesis that CA1 ensemble dynamics over timescales of days carry information about the temporal relationship between experienced events, both within and across spatial contexts. We wish to elucidate that unlike recent studies that introduced the imaging technique that we have used here (Ziv et al., 2013, and Ghosh et al., 2011), the current work is not about the technique. Several conceptual aspects of our work prove it to be a substantive advancement beyond the methodology:

1) Distinct CA1 representations co-evolve with time.

We agree that the idea that changes in neuronal activity could infer time has been investigated in multiple studies. For example, Manns et al., 2007 showed population evolution at a time scale of minutes and Mankin et al., 2015 showed dynamics in CA2 that allowed the distinction of representations for up to 6 hours (CA2 was shown to reach a plateau at longer time intervals). However, unlike previous studies, in this work we focused on the co-evolution of CA1 representations of different contexts. Until now it has not been explicitly demonstrated that CA1 neural codes can be both substantially different between contexts and yet co-evolve with time in a manner that allows the binding in memory of events that occurred at different contexts. In the revised manuscript we further substantiate that the CA1 representations of the two environments are highly distinct (Figure 1, Figure 1—figure supplement 3 and Figure 2—figure supplement 6). (This is an important point because it would not be surprising if representations that were similar to each other were found to co-evolve with time). We also show that our time decoders could infer the day from which the test data was taken – both within and across environments – even if we use data from only place cells or from only non-place cells, suggesting that time stamping is not intrinsically linked to specific aspects of the episodic representation such as place coding. Overall, the results support our hypothesis, and suggest that CA1 ensemble dynamics can both associate or dissociate distinct experienced events in long-term memory based on their temporal distance.

2) Days-scale dynamics are relevant to long-term episodic memory.

Whereas previous electrophysiological studies have attributed a role in time coding for changes in hippocampal neuronal activity over timescales of seconds to ~1.5 days, our study investigated CA1 coding dynamics over much longer timescales of days–weeks. Dynamics over such timescales are relevant for long-term episodic and autobiographical memory, and specifically for the ability to form a mental timeline of events that were experienced days or weeks apart. By analogy, just as recordings of place cells in simple small-sized environments can only inform us to limited extent about the way larger real-life environments are represented, recordings over minutes–hours may not reveal all aspects of the dynamics that are relevant to coding of time in long-term memory. Our results show that the hippocampus could enable the formation of a timeline of memories over extended days-weeks timescales while maintaining coding specificity for different environments, consistent with its hypothesized role of encoding ‘where’ and ‘when’ into long-term memory.

3) “Time decoders” quantify time information carried by the ensemble dynamics.

Whereas previous studies have averaged the data over mice and/or recording days, the large-scale recordings in our study allowed us to longitudinally analyze the data at the ensemble level, and for each mouse individually. To quantify the degree to which the ensemble dynamics is informative about time, we devised three different approaches for neuronal “time decoding”. Using these analyses we demonstrate that events that occurred at temporal proximity (e.g. the same day) shared a common timestamp, even if these events have occurred in two distinct spatial contexts. We also found that information about time is embedded in the ensemble activity even in short, seconds-long fractions of a session. These are new results, which could not have been derived from previous studies that characterized the dynamics of modest numbers of neurons per animal. Furthermore, thanks to suggestions by the Reviewers, we now provide a detailed analysis of the dynamics of individual cells, and of different subsets of cells. The new analyses show that weak monotonic changes in firing rates at the single cell level can translate into strong monotonic changes at the ensemble level (Figure 2—figure supplement 1), and that both place cells and non-place cells contribute information about time (Figure 2—figure supplement 4).

Reviewer #2:

*The report by Rubin et al. finds that hippocampal CA1 ensembles gradually change over time such that ensemble that are recorded several days apart do not resemble each other. In contrast, they find that CA1 ensembles that are recorded on the same day in two different environments resemble each other. While the number of cells that can be recorded with calcium imaging is certainly much higher than with standard electrophysiological techniques, the reported results largely resemble those that were obtained in previous recording studies which have already reported and proposed many of the key conceptual points. Notably, closely related publications (e.g., Manns et al., 2007; Howard and Kahana, 2002, and others) are not included here and standard analysis methods (detailed comparison of firing during immobility and across the two environments) are not performed.*

We agree that our findings are consistent with previous electrophysiological studies in the field, and recognize that our writing did not sufficiently emphasize the novelty of our results with respect to these studies. We hope the revised manuscript now clarifies this issue. Thank you also for drawing our attention to the studies by Manns et al. (2007) and Howard & Kahana (2002) which are now cited in the Introduction.

*1) The Introduction begins with mentioning the work of others, but in the second half, predominantly cites the work from their own group (Ziv et al., 2013). Furthermore, it is not mentioned that Mankin et al. (2012 and 2014) recorded in two environments and reported correlations between the two environments, which show a similar pattern as those reported here.*

We now cite the studies by Manns et al. (2007) and Howard & Kahana (2002) in the Introduction, and in the Discussion mention that Mankin et al. (2012 and 2015) recorded in two environments. The revised manuscript now states:

“Pyramidal neurons in the hippocampal area CA2 (upstream to CA1) were found to display representations that are similar for different contexts but highly variable upon repeated exposures over timescales of hours. It was suggested that such CA2 dynamics, when combined with spatial information from CA3, could jointly reflect spatial and temporal information in CA1 (Mankin et al., 2012, 2015).”

*2) Paragraph four. Non-place cells are presented as if it were not a well-known fact that many cells that are not place cells are active during sharp-wave ripples and in other phases of the task. In fact, many well performed electrophysiology studies report the fraction of cells that are place cells compared to those that are only observed during immobility.*

Our intention was not to imply that it was not well known that non-place cells are also active during different phases of the exploration task. The aim was to show that in our data non-place cells generated nearly half of the Ca^2+^ transients; but by no means do we claim this to be a new finding. We omitted the words “typically analyzed” from the text to avoid giving this impression. The revised manuscript now states:

“Thus, neuronal activity during a training episode on the linear track was much richer than the part of the data that is limited to place codes.”

*A much more detailed analysis of the firing patterns of non-place cells within and across environments is needed before any conclusions about their contribution to stability over time can be drawn.*

Thank you for this suggestion. We now provide a new supplementary figure (Figure 2—figure supplement 4) with a detailed analysis of the temporal information carried by place cells and non-place cells. First, we compared the performance of the time decoders on Ca^2+^-event data from epochs in which the animal was either moving or stationary (mostly at the ends of the tracks). The performance of the time decoder on these two parts of the data was similar. Next, we compared the performance of time decoders that were trained on data from either place cells or non-place cells. Both subsets of cells supported a highly significant performance of the decoders, and were not significantly different from each other.

*3) Paragraph five. A much more detailed analysis of the correlation across environments for place cells and non-place cells is needed. It is of course expected that correlations within environments are higher than between environments but the key question is the extent of the neuronal coding difference for the two environments.*

Thank you for suggesting this. We now added Figure 2—figure supplement 6, which presents the correlations of the place cell population vector within and between the two environments. The correlation values between environments were low throughout the experiment. Thus, at any point along the course of the experiment the place cell representation of the two environments was different. As mentioned in the previous point, Figure 2—figure supplement 4 shows that both place cells and non-place cells carried information about time.

*From the information that is currently provided is appears that A and B are partially correlated and that each environment is more correlated with itself from at temporally proximal compared to temporally distant time points. If A and B and A and A' are correlated, it appears trivial that B and A' are also correlated, and all classifiers and conclusions seem to be based on this finding.*

If we understand correctly the notation you use (i.e. A and B are the representations of the two environments, and B’ is the representation of the same environment as in B at a different time), our data suggest that the correlation between A and B is higher than the correlation between A and B’. This result could not have been derived from the correlations between A and A’, or B and B’ alone. To place this logic in perspective, consider that previous work showed that the place code for a fixed environment changes over timescales of days–weeks. But what has remained unknown until our current study is the degree to which such days-scale evolution in the hippocampal CA1 representation of a given environment is independent from the evolution of the representations of other environments. We think this point is far from trivial: if the dynamics of the representations of different environments were independent, then different visits to the same environment would still be encoded as different traces; however, with such a coding scheme events that occurred at different environments but at temporal proximity will not be associated in long-term memory. Contrary to this scenario, our current work revealed that the representations of two different environments co-evolved with time, sufficing to infer the time in which events occurred in one environment using reference data from the other environment.

*4) Paragraphs six and nine. These results have been shown before by others.*

The reviewer refers to the results of the ordinal time decoder, which indicate that day-to-day differences in activity patterns of CA1 neurons carry ordinal information. We accept that our findings are consistent with previous work that showed changes in CA1 neuronal activity over seconds-hours (Manns et al., 2007), hours-days (Mankin et al., 2012) and days-weeks (Ziv et al., 2013), and now state this explicitly in the revised manuscript. However, to the best of our knowledge, none of our ensemble-level time decoding approaches regarding the degree to which days-scale ensemble dynamics carry information about time have been published before. On this point the revised manuscript now states:

“Our present findings of coding dynamics over timescales of days are consistent with a key feature of episodic memory— that each experience is uniquely encoded. […] This idea is also consistent with previous reports of changes in CA1 neuronal activity over timescales of seconds-hours (Manns et al., 2007), hours-days (Mankin et al., 2012) and days-weeks (Ziv et al., 2013).”

*5) Paragraphs eleven and thirteen. Previous similar findings are not discussed.*

We have revised the text and now cite and discuss previous similar work (see points #4 above, and #7 below).

*6) Paragraph twelve. The largest fraction of change is found within 1-2 days, which is not consistent with the time course of neurogenesis.*

Indeed, the large fraction of the change is within 1-2 days, but results from the current study show a gradual change over the course of weeks. Thanks to the reviewers’ suggestions, we now show that this gradual change is sufficient for the performance of the time decoders (Figure 2—figure supplement 3). Furthermore, there is currently no evidence at the neural code level regarding the degree to which neurogenesis in the DG affects representational stability in the downstream CA1. However, there is relevant evidence at the behavioral level: neurogenesis rates were found to be inversely correlated with the strength of long-term episodic-like memories, or the duration of such memories were hippocampus dependent (Akers et al., Science 2014, and Kitamura et al., Cell 2009). Given that the CA1 is the main output from the hippocampus, it is plausible that such behavioral effects were associated with representational differences in CA1.

*In contrast to what is reported here (i.e., constant remapping between environments), Rangel et al. (2014) found a change in the degree of remapping over the time course of neurogenesis and when manipulating neurogenesis.*

We do not think our results are inconsistent with those reported by Rangel et al. In their work, cells exhibited higher context selectivity if the rats were introduced to the different contexts gradually over weeks (first context 1, then context 2, and finally context 3), than if exposed to all contexts in the same week. Our experimental design is very different: we pre-trained the mice for 8-11 days in both environments, before starting the two-week long imaging experiment, but did not examine the effects of gradual versus simultaneous introduction of the two environments on their representations or dynamics. Beyond this basic conceptual difference in study design, there are multiple additional differences between the studies that make a direct comparison between them difficult. For example, in Rangel et al., they recorded from the DG of rats whereas we recorded from the CA1 of mice. The DG and CA1 have different qualitative and functional differences. Rangel et al. pooled data from four days of recordings at the end of the training phase, whereas we recorded longitudinally over two weeks. They did not compare neuronal activity from the same animals at different intervals along the course of the experiment, whereas we did. In their experiments rats explored environments of different sizes, whereas in our experiments the environments were the same size. In their experiments distal external cues were the same for the different environments, whereas in our experiment they were different.

*7) Paragraph thirteen. While the experimental design in Mankin et al. (2012) could distinguished between effects of elapsed time and intervening experience, no such results are reported here. If this is discussed, the text should refer to previously published data on this topic.*

We agree with this comment and have now added a sentence to the Discussion that makes reference to this previous work. The revised manuscript now states:

”Previous work found no effect for the number context exposures on the rate of change in CA1 codes over 30 hours (Mankin et al., 2012), but it is possible such an effect could be observed over longer timescales.”

Reviewer #3:

*1) The Abstract exclusively emphasizes the time evolution of CA1 representations independent of spatial context. The absence of a clear statement in the Abstract that important aspects of the representations of spatial contexts remain constant over time – the preserved locations of place fields (already in the Ziv et al., 2013 Abstract) and the preserved global remapping between spatial contexts (new here) – could lead to misunderstanding by readers. Therefore, the authors should modify the Abstract so that it is clear which important aspects of the representation of spatial contexts remain stable over long time periods.*

Done. We have revised the Abstract, and added the information about the important aspects of the representation that remains stable over time. The new sentence states:

“Longitudinal analysis exposed ongoing environment-independent evolution of episodic representations, despite stable place field locations and constant remapping between the two environments.”

*2) The ability to order representations of each environment across days could be inferred from Ziv et al. (2013). What would have been the alternative to this, given the findings in this previous work? (On the other hand, the stability of global remapping across time and the co-evolution of activity levels across multiple environments is new.)*

The results of previous work indeed hinted that we should be able, given similar data, to order the representations of each environment. However, given that the analyses in Ziv et al. (2013) are based on data averaged over four mice, and in most cases also over day-pairs, it has been unclear to what degree ordering the representations will be accurate at the individual mouse and day levels. The gradual changes that were observed when averaged over all mice could originate from abrupt changes in the representations in individual mice. Such abrupt changes in the representations of each environment would not support perfect ordinal sorting of the data. Furthermore, taken together with our findings of the co-evolution of the representation of the two environments, we have now ruled out the possibility that the previously reported results regarding time evolution of a representation of a given environment are only due to dynamics in the representation that are specific to that environment.

*The authors should delve into the details of the time evolution of single-environment representations more. For example, is the gradual evolution of activity levels over time due to half of the neurons gradually increasing their activity levels over the 2 weeks and half decreasing? Or do many neurons increase then decrease, or decrease then increase, activity over that period? Or does some substantial fraction of cells have activity levels that remain basically the same over time? What are the percentages of each kind of pattern?*

We now added a new supplementary figure (Figure 2—figure supplement 1) that presents a detailed analysis of the cell-level time evolution in a single environment. Specifically, we found that:

1) There were no cells that exhibit monotonic increase or decrease in activity throughout the two-week period of the experiment.

2) Nevertheless, the single-cell dynamics were slightly more monotonic than the shuffled data. Such weak monotonic behavior at the single cell level seems to underlie monotonic dynamics at the ensemble level.

3) Even neurons that were active throughout all eight days of the experiment did not tend to demonstrate a constant firing rate.

*3) For the within-environment decoder, it seems not surprising that the activity pattern in a trial of a day would best correlate with the activity pattern on the rest of that day. The important issue is whether the pattern evolves gradually over time versus just being noisy over time. Therefore, what the authors should do is remove the session from the day of the test trial and see if the best match from the remaining days came from a day that was closer to the actual day than chance. Similarly, this calculation should be done for the across-environments decoder (that is, eliminating the same day's data). (However, it is useful to also have the answer when the same day is included, which is the calculation they already did.)*

Thank you for this suggestion – we agree that it is not surprising that the activity patterns in a trial of a session would best correlate with the activity pattern of the rest of the session. We have now added these analyses in a new supplementary figure (Figure 2—figure supplement 3). The results confirm that the ensemble activity evolves gradually over time.

*4) An important part of the memory function of the hippocampus would be the ability to distinguish between the two spatial contexts and correctly identify them with previous experience in each of those contexts. While the authors showed intact global remapping between environments within each day, they should analyze if they can distinguish which previous environment they are recalling across days. Specifically, they should compare how the spatial population vectors of the representation of environment A on day n for n = 3, 5, […] correlate with the spatial population vectors for environment A and separately for environment B on day n = 1. Then they should repeat it for day 5, 7, […] versus day 3, and so on. (This should also be done for environment B versus environment B or A on different days.) This analysis will essentially be an across-days combination of Figure 1 and Figure 2.*

We have now added these analyses in a new supplementary figure (Figure 2—figure supplement 6). The results confirm that at any point throughout the experiment the ensemble representation of the two environments was different.

*5) The authors write "Thus, having shown that remapping between familiar environments is stable over time, we tried in our subsequent analyses to account for the entire episodic representations. We therefore opted to include in the following analyses all the active cells and their Ca^2+^ events, regardless of their spatial running properties." They describe in the Materials and methods that they excluded the 2 spatial bins on each end of the tracks when defining place fields, but did they then include these bins when determining the spatial population vector correlation between representations in different environments to show remapping? And I assume based on the quoted text above that they did not exclude the 2 spatial bins on each end of the tracks when defining the "ensemble activity patterns" for each time period, which was used to do their time evolution analysis? This should be clarified in the Materials and methods and main text.*

Yes, the last two bins of the tracks were excluded from the analysis of the population vector correlations (Figure 1), but included in the analysis of the ensemble activity patterns (Figure 2). This is now clarified in revised manuscript.

The caption for Figure 1 now states:

“(G) Average correlations of place cell firing patterns (i.e. the ‘population vector’, PV) during running epochs at different locations along the two linear tracks were low and did not change over time”

The Materials and methods section now states:

“For each spatial bin (excluding the last 2 bins at both ends of the tracks) we defined the population vector as the mean event rate for each neuron given that bin’s occupancy.”

*6) Related to (4): They appear to have excluded the reward-related activity at the track ends in the global remapping analysis. They appear to have included this activity in the "ensemble activity pattern" vectors – which contain the relative activity of each cell in a given environment, independent of the locations at which that activity for each cell occurred – for the time ordering analysis. Assuming this is the case, the authors should re-run the analysis for the time ordering results using only the subset of cells that show place activity and, separately, all cells excluding the track ends where reward is consumed because the remaining activity would represent the theta-state associated activity patterns. This analysis would also reveal to what extent the reward-related and other activity at the track ends contributes to their time ordering results.*

Thank you for this suggestion. To quantify whether non-place codes contained temporal information we analyzed the performance of the different time decoders, separately for periods in which the mouse was active versus stationary (Figure 2—figure supplement 4), and for place cells versus non-place cells (Figure 2—figure supplement 4). Active periods were defined as times when the animal was at velocity ≥1cm/sec and located more than 8cm from either end of the track. Place cells were defined as cells with significant place coding in at least one session. Figure 2—figure supplement 4 now shows that: (1) Ca^2+^ events taken from epochs in which the mouse was either active or stationary, and (2) place cells and non-place cells all carried significant temporal information. Thus, time stamping is not intrinsically linked to specific aspects of the episodic representation such as place coding.

*7) One important difference between this study and the previous one (Ziv et al., 2013) is the use of GCaMP6s and GCaMP6f here versus the previous use of GCaMP3. Both studies point out that there is significant change in the hippocampal representation over time. However, the GCaMP6 variants are more sensitive detectors of neural activity than GCaMP3, which detects strong bursts more than single spikes. Because models of memory can depend on parameter values such as how stable activity is over time, it would be very valuable for the hippocampal field to have a quantitative comparison between the key numbers from the previous GCaMP3-based study and the GCaMP6-based values here (especially since other differences in methodology are likely to be low due to overlap in the author lists). Therefore, the authors should quantitatively compare their data to Ziv et al. (2013) by adding to the current manuscript's supplementary figures the same plots corresponding to Figure 2, Figure 3B, and Figure 3C of Ziv et al. (2013). It looks like one of these (the fraction of cells that are place cells) is already shown in both (Ziv et al., 2013 Figure 2 that shows a place cell fraction of ~10-15% and this manuscript's Figure 1 shows a value of ~50%), but it's not clear that it has been computed in exactly the same way. As for Figure 3C in Ziv et al. (2013), the previous study used a binary distinction between active and inactive cells. The current manuscript's Figure 2 uses a more graded activity level measure for cells. In the inset of Figure 2, does the "overlap" plot use exactly the same definition of active and inactive cells as Ziv et al. (2013) Figure 3C? Finally, the authors should show separate plots for GCaMP6s and for 6F, which could have different detectability for events.*

We have now added a new supplementary figure (Figure 1—figure supplement 4) that addresses these issues, and separately presents for GCaMP6s and GCaMP6f the analyses done in Ziv et al. (2013) for: (1) number of days that the cells were active in each environment, (2) fraction of place cells active during rightward and leftward movement in each environment, and (3) the changes in the ensemble activity correlations, or overlap, within and between environments. Please note that while we present here data that was analyzed as in Ziv et al. (2013), several other aspects of the current experiment (beyond the use of a more sensitive indicator) were different than the previous studies. For example, the interval between imaging days was five days in the previous study and two days in ours, which may account for some of the differences seen in the overlap between the place cell ensemble that were active on different days; in the previous work mice explored a single environment, whereas in the current study mice explored two different environments in each day of recording; both the length of the linear tracks and their distance from distal visual cues are longer in the current study, which may have contributed to some of the differences in the fraction of place cells, or the degree to which they are direction selective.

*8) How does the data relate to Lever et al. (2002)? In that paper, it was shown that the representations of the two environments diverged over time. Here, did the representations of the two environments separate more with time in certain aspects? While the authors show that their global remapping score is stable over time, it looks like Figure 2 shows some degree of increasing divergence even though Day 1 already represents ~2 weeks of familiarization with both environments. The authors should check for this and report any change or lack of change.*

As suggested, we adopted a similar approach to that presented in Lever et al. (2002) and added two new analyses of the degree to which the representational difference between the two environments changes with time (Figure 1—figure supplement 4). Due to the nature of the Ca^2+^ imaging data, in which bursts of action potentials are often detected as single Ca^2+^ excitation events and not as trains of spikes (as in electrophysiological recordings), it is hard to reliably evaluate a spatial activity map of single neurons based on a single 3-minute trial. By comparison, Lever et al. typically analyzed 8-10 minute trials. We therefore devised two scores: ‘activity divergence’ and ‘peak displacement’, which resemble Lever’s ‘rate divergence’ and ‘peak divergence’ scores, but different in that they do not require the evaluation of firing map based on data from a single trial. Importantly, according to our new analyses the dynamics of neither of these scores changed over the time scale of the experiment.